# Understanding Mental Health Needs and Gathering Feedback on Transcutaneous Auricular Vagus Nerve Stimulation as a Potential PTSD Treatment among 9/11 Responders Living with PTSD Symptoms 20 Years Later: A Qualitative Approach

**DOI:** 10.3390/ijerph19084847

**Published:** 2022-04-16

**Authors:** Rebecca M. Schwartz, Pooja Shaam, Myia S. Williams, Molly McCann-Pineo, Laura Ryniker, Shubham Debnath, Theodoros P. Zanos

**Affiliations:** 1Department of Occupational Medicine, Epidemiology and Prevention, Northwell Health, 175 Community Drive, Great Neck, NY 11021, USA; rschwartz3@northwell.edu (R.M.S.); mmccann1@northwell.edu (M.M.-P.); lryniker@northwell.edu (L.R.); 2Institute of Health System Science, Feinstein Institutes for Medical Research, Northwell Health, 350 Community Drive, Manhasset, NY 11030, USA; mwilliam26@northwell.edu (M.S.W.); tzanos@northwell.edu (T.P.Z.); 3Center for Traumatic Stress, Resilience and Recovery, Northwell Health, 175 Community Drive, Great Neck, NY 11021, USA; 4Donald and Barbara Zucker School of Medicine at Hofstra/Northwell, Northwell Health, 500 Hofstra University, Hempstead, NY 11549, USA; 5Department of Medicine, Northwell Health, 600 Community Drive, Manhasset, NY 11030, USA; 6Institute of Bioelectronic Medicine, Feinstein Institutes for Medical Research, Northwell Health, 350 Community Drive, Manhasset, NY 11030, USA; sdebnath@northwell.edu

**Keywords:** posttraumatic stress disorder, PTSD, vagus nerve stimulation, taVNS, focus group, world trade center, 9/11, focus group, first responders

## Abstract

Posttraumatic stress disorder (PTSD) remains one of the most prevalent diagnoses of World Trade Center (WTC) 9/11 responders. Transcutaneous auricular vagus nerve stimulation (taVNS) is a potential treatment for PTSD, as it can downregulate activity in the brain, which is known to be related to stress responses and hyperarousal. To understand barriers and facilitators to engagement in mental health care and the feasibility and acceptability of using the taVNS device as a treatment for PTSD symptoms, a focus group was conducted among patients from the Queens WTC Health Program who had elevated symptoms of PTSD. The focus group discussion was recorded, transcribed, and analyzed. Three themes and subthemes emerged: (1) the continued prevalence of mental health difficulties and systematic challenges to accessing care; (2) positive reception toward the taVNS device as a potential treatment option, including a discussion of how to increase usability; and (3) feedback on increasing the feasibility and acceptance of the research methodology associated with testing the device in a pilot clinical trial. The findings highlight the need for additional treatment options to reduce PTSD symptoms in this population and provide key formative phase input for the pilot clinical trial of taVNS.

## 1. Introduction

The 9/11 World Trade Center (WTC) disaster inflicted lasting psychological trauma on survivors, witnesses, and first responders throughout New York City and beyond [1]. An estimated 40,000 to 60,000 first responders who provided emergency services at Ground Zero were exposed to various environmental contaminants and suffered from a variety of physical and mental health conditions [2]. Posttraumatic Stress Disorder (PTSD) remains the most prevalent mental health diagnosis for first responders and clean-up workers at Ground Zero, and rates of PTSD symptoms continue to be higher than rates in the general population [3]. Previous research has estimated that as many as one-fifth of WTC responders met the screening criteria for PTSD in the aftermath of the exposure, and that persisted for years such, that 9.7% of WTC responders who were interviewed 11–13 years after 9/11 met the criteria for current PTSD [4,5].

Current clinical practice guidelines for the management of PTSD recommend trauma-focused psychotherapies, including Cognitive Processing Therapy (CPT), Prolonged Exposure (PE), and Eye Movement Desensitization and Reprocessing (EMDR), as first-line psychotherapeutic interventions to engage with the index trauma and the associated fears and cognitions [6]. While these evidence-based treatments (EBTs) are effective and empirically supported, there are challenges associated with their implementation among military and veteran populations and are only marginally more effective than non-trauma-focused treatments in that population [7]. Given the lack of clinical trials focused exclusively on WTC responders, the efficacy in such a population is unclear, but there is reason to believe that first responders are similar to the military and veteran population in terms of PTSD symptomatology and barriers to engagement in treatment [8].

Despite the substantial progress made in evidence-based pharmacological, psychotherapeutic and behavioral interventions, stigma, lack of knowledge regarding viable treatment options, and financial and time constraints have been reported barriers to engagement in care for responders with PTSD [9]. Furthermore, complementary and alternative medicine (CAM) approaches, which include a range of therapies, from yoga to acupuncture to neurostimulation, are not considered to be standard in the current practice of Western medicine but have modest beneficial effects as a treatment for PTSD [10]. While the VA supports efforts to study the benefits of CAM for PTSD, there remains a clear need for new PTSD treatments that are effective for and amenable to the unique circumstances of veterans.

The World Trade Center Health Program (WTCHP), which was established in 2002 to address the physical and mental health needs of first responders, published a recent report which indicated that in 2017, about 5% of continuously enrolled WTCHP responders utilized mental health services (excluding pharmacy) [11]. Among WTCHP responders with a WTC-certified mental health diagnosis, only about 55% utilized mental health care and 99% of these services were delivered in an outpatient setting [11].

Even a less demanding approach to addressing PTSD, such as the use of psychopharmaceuticals, demonstrates varying efficacy when used without psychotherapy treatments [12]. A plethora of treatment-related factors, such as treatment type and modality, insufficient time with the mental health professional, treatment ineffectiveness, work interference, personal problems, or discomfort with medical professional interactions, have been found to impact treatment dropout rates [13,14]. Thus, given the high rates of PTSD and the lower uptake of effective PTSD treatment among responders, there is a clear need to provide practical and easily accessible treatments to WTC responders to alleviate PTSD symptoms [7].

Exposure to events or stressors, primarily those that are similar in nature to previously experienced trauma, can elicit symptoms such as hyperarousal, intrusive thoughts, avoidance behaviors, and dissociation, which can lead to elevated inflammatory marker concentrations, impaired autonomic modulation, memory deficits, changes in brain morphology, and increased neural reactivity in emotion-specific brain areas [13]. Neurostimulation techniques such as electroconvulsive therapy (ECT), repetitive transcranial magnetic simulation (rTMS), and vagus nerve stimulation (VNS) have been utilized to address various mental health difficulties with varying levels of invasiveness and potential side effects [15].

Vagus nerve stimulation (VNS) can potentially benefit treatment of psychiatric disorders partly due to its projections to brain areas such as the amygdala and hippocampus by downregulating activity in these areas which are known to be related to stress responses and hyperarousal [16]. Therefore, VNS is a potential treatment method for PTSD as it modulates sympathetic tone and related cardiovascular reactivity, and it has even been shown to enhance fear extinction in rodents trained with a conditional fear paradigm [13]. One study found that among rats that were subjected to a single prolonged stressor, impairment in the fear extinction response was reduced, as was the subsequent conditioned fear response, when VNS was administered as compared to a sham [17]. In addition, PTSD-like symptoms, such as anxiety, hyperarousal, and social avoidance, were reduced in the VNS animals in the short term and for more than 1-week post-VNS [17].

Although studies are limited, research involving human subjects has also found support for the use of VNS as an adjunct therapy to address treatment-resistant depression and anxiety with no adverse events [18]. Bremner et al. (2019) found decreased inflammatory markers, decreased sympathetic tone (increased tone is associated with increased stress), and increased medial prefrontal function with VNS as compared to sham controls [19]. Further studies using VNS either transcutaneously or using implanted devices have shown that it has even been effective in treating diseases such as tinnitus, atrial fibrillation, episodic migraine, seizure frequency, cluster headache, and major depression and was even shown to improve vagal tone, to deactivate limbic and temporal brain areas, and to improve mood in young and healthy as well as older individuals [13]. These results point to the potential efficacy and safety of VNS to address PTSD symptoms in individuals who have not accessed or engaged with traditional EBTs, such as WTC responders. As such, it is necessary to understand the acceptability and feasibility of using a novel, safe, and noninvasive form of VNS, transcutaneous auricular VNS (taVNS), which consists of a neural interface ear piece that generates electrical pulses delivered transcutaneously to the auricular branch of the vagus nerve to address PTSD symptoms in WTC responders.

The current study involved the implementation of a formative phase evaluation in which a focus group was conducted with WTC responders with elevated PTSD symptoms. The aims of the focus group were to understand responders’ perspectives on (1) the mental health needs of WTC responders as well as barriers and facilitators to engagement in mental health care and (2) the feasibility and acceptability of using the taVNS device as a treatment for PTSD symptoms as well as the pilot study methodology. The latter allowed the study team to tailor the taVNS intervention and the pilot study methodology so that it is relevant and acceptable for use with WTC responders with PTSD in order to then test VNS efficacy robustly in a future randomized controlled trial (RCT).

## 2. Materials and Methods

### 2.1. Participant Recruitment

The recruitment goal was to receive consent from and enroll 10 participants from the Queens World Trade Center Health Program (WTCHP) to participate in the focus group, which was held at the Queens WTCHP site. Participants were eligible to participate in the focus group if they were (1) a WTCHP responder who agreed to be contacted to participate in research; (2) diagnosed as having PTSD, as per the WTCHP certification criteria as indicated by the WTC General Responder Data Center; and (3) a responder who had elevated symptoms of PTSD, as per the PTSD Checklist–Specific (to 9/11) (PCL-S) [20] score of ≥44 during an annual monitoring visit between 2018–2020. Responders physically or mentally unable to consent or participate, or those unable to speak, read, or write in English, were excluded. For the purpose of this study, investigators received a list of those eligible for recruitment, which included a total of 78 patients, based on the three inclusion criteria from the Queens WTCHP, and the list was randomized using the RAND syntax in Excel. A research coordinator sent out an email to all 78 patients on the list and contacted 50 of the patients by phone in the randomized order until 10 people who met the eligibility criteria agreed to participate in the focus group. Six participants attended and provided informed consent before the moderator started the focus group, which was conducted in October 2021 (Figure 1). Two participants were no-shows despite receiving reminder emails and a phone call prior to the focus group, while two participants reached out to us indicating a work conflict that prevented them from attending the focus group. Participants were made aware that the focus group session was voluntary and that they could withdraw at any time. To protect the identity and confidentiality of participants, pseudonyms were used (e.g., Participant 2), and any identifying details were adjusted as needed.

This study was reviewed and approved by the Institutional Review Board with exempt status.

### 2.2. Focus Group Procedure

The focus group was facilitated by one member of the research team with a secondary co- facilitator and a note-taker, all from the study team. A semi-structured focus group discussion guide was utilized, composed of open-ended questions that were informed by the research focus and aims and guided by previous research. All study investigators contributed feedback to the development of the focus group guide. Some example questions included “*Can you please describe what you see as WTC responders’ greatest needs currently as they relate to mental health*?” and “*What are your initial impressions of the [taVNS] device itself?*” (see Appendix A for the full discussion guide.) Study team members were cognizant that discussing 9/11 events may have generated additional stress for the participants, so the primary focus was inquiring about their perceptions of WTC responders’ mental health needs in general as well as of the taVNS intervention itself, not details regarding their previous 9/11 traumatic experiences. The Queens WTCHP Director of Mental Health was in close communication with the study Principal Investigator should any participants experience distress during the focus group, and his information was also made available to all focus group participants. The focus group was approximately 60 min long, and participants were compensated $50 for their participation and travel reimbursement.

### 2.3. Qualitative Analysis

The focus group discussion was audio-recorded and professionally transcribed verbatim. An inductive thematic analysis [21] was used to analyze the transcribed focus group discussion. To improve credibility, dependability and transferability of results, peer debriefing, documentation of results, and researcher triangulation were employed, following the six phases set forth by Braun and Clarke [21]. Two members of the research team independently read the transcripts several times to capture the overall perspectives of the participants (Phase 1). These two investigators focused on patterns in the data to generate an initial set of codes (Phase 2). Next, they sorted and collated the codes into potential themes (Phase 3) and then reviewed the themes by first determining any coherent patterns and then determining whether those patterns were aligned with the specific aims (Phase 4). For Phase 5, the investigators “defined and refined” their themes and then met to discuss their findings. All potential themes and subthemes were discussed by the two investigators and revised when necessary. Additionally, disagreements were rectified through open discussion and revised until consensus was reached by both investigators. Lastly, a report was generated (Phase 6), and all members of the research team met and discussed the themes to ensure that they reflected the participants’ perspectives and were in alignment with the purpose of the research aims.

## 3. Results

### 3.1. Sample Characteristics

Out of the 10 responders who agreed to participate in the focus group, 6 actually attended. Three participants identified as male and three as female. Their ages ranged from 51 to 77 years old. The average PCL score of the six participants at their last monitoring visit was 56, with a range between 45 and 70, indicating a level above the clinical threshold for PTSD among the participants who attended the focus group.

### 3.2. Emergent Themes

Results are presented as three overarching themes with subsequent subthemes that depict the perspectives of participants with the goals of identifying mental health needs as well as barriers and facilitators to mental health care engagement and ensuring that the taVNS device and the pilot study methodology are relevant and acceptable for use with 9/11 WTC responders with PTSD (Table 1).

#### 3.2.1. Theme 1: Mental Health Needs; Barriers/Facilitators to Engaging in Care of 9/11 Providers

The first theme shows the mental health needs and barriers to mental health care engagement of 9/11 WTC responders with PTSD. Participants reported that, in addition to PTSD, they also suffered from other mental health disorders (e.g., anxiety), which can be associated with having PSTD. Despite having documented mental health difficulties, participants indicated that there were systemic challenges to getting access to treatment for their psychological concerns. One such recurring challenge that was reported was the perception that their mental health is not taken seriously by government officials nor their employers. Participants also provided recommendations as to how barriers can be addressed.

##### Sub Theme 1: Mental Health Difficulties Continue to Be Prevalent

The participants spoke about the other mental health challenges that they have developed over the past 20 years in addition to PTSD. The frequently mentioned challenges included anxiety and sleeping problems. Participants also underscored the fear and avoidance associated with PTSD. One participant noted, “*I don’t even ride a train because of anxiety. I get…I get fear…I start sweating. I can’t board the train*.” Another participant commented on the pervasiveness of the disease by saying how, even years after 9/11, their symptoms have only gotten worse: “*And as**…as time goes by, it gets**…it gets worse. It doesn’t get any better. My problem didn’t start for a few years after, and then when it**…when it started, the**…the sounds and**…and it was so horrible and so loud, and I hear the…from that and it gets louder and louder and I have to get up. I can’t sleep.*”

##### Sub-Theme 2: Systemic Challenges to Getting Access to Treatment for PTSD and Other Mental Health Problems

Participants reported that they faced a number of challenges with getting their mental health needs addressed. Some of those challenges included stigma (e.g., “*There is a stigma with mental health in general, but there’s a stigma at least from my standpoint in the law enforcement*”), retaliation (e.g., “*the human resource department of my agency actually used my confidential FMLA package and sent me to a city psychiatrist and turned the PTSD against me*”), lack of taking PTSD and other mental health challenges into consideration (e.g., “*… and when I applied for Victim Compensation Fund, went through the whole process, they told me**…the lawyer said you can’t even put in anything about PTSD because mental health is not even being taken into consideration*”), and lack of access to and availability of viable treatment options (e.g., “*So, the other**…one other thing is I think that they don’t have enough things available for us to do besides give us drugs and talk therapy*”). In addition to the systemic challenges that participants reported, they also indicated other barriers to receiving treatment such as transportation difficulties, lack of access to treatment facilities, and perceptions of bias in ability to receive treatment based on factors such as age.

##### Sub-Theme 3: Suggestions to Overcome Barriers to Mental Health Treatment

Participants provided some suggestions that they felt could be useful in helping reduce the barriers associated with addressing mental health needs. It should be noted that participants were in unanimous agreement to move from talk therapy and traditional forms of treating mental health to more holistic approaches. One participant said, “*….but there’s other things that, you know**…it’s traditional medicine, but what about acupuncture which has been**…I mean, if you look at the studies that they’ve done with military and with other certain groups, you can see that there are some**…certain holistic things that have worked; yoga, meditation, these kinds of things, and why not make those things available too, salt caves*.” Additionally, participants reported earlier that transportation was a barrier to access to mental health; hence, one participant suggested a potential solution, “*Transportation voucher…or some… yeah, something where like you**…you maybe deal with the parking*.”

#### 3.2.2. Theme 2: Device Feedback

Participants were given an opportunity to view and handle the taVNS device and to view and comment on the instructional video and accompanying materials associated with the device. Overall, the initial reactions were unanimously positive. Participants also had a few questions about the device, most of which focused on concerns regarding the safety of the device when used by people with specific health issues.

##### Sub-Theme 1: Initial Reactions

The initial reactions towards the device itself were overwhelmingly positive, with some participants expressing enthusiasm for the novelty of the taVNS as a potential treatment for PTSD. Some of the reactions were: “*I think it’s great that… I just think the fact that they are doing something that has to do with technology and they’re**…to me it means like somebody is really making an effort, you know, to do something*” and “*Okay, I really love it because I’m tired of taking medicine*.” Another participant commented on the convenience and transportability of the device by saying, “*… So, I think that’s a big barrier is when you have to go to somebody’s office, and you have to travel and you have to get off the work, you have to**…like I could do it on a lunch break, I can, you know, you can really do it anywhere and that’s huge, huge*.”

##### Sub-Theme 2: Concerns about Interactions with Other Illnesses or Comorbidities

Because of the device’s underlying mechanism of nerve stimulation, some participants were concerned that it may affect previous/current illnesses and comorbidities, such as stroke (e.g., “*I had a stroke in the past, I have seizures. Can I use it? Isn’t that a problem?*”) and tinnitus (e.g., “*Does this have any**…does this make tinnitus worse at all?*”). Investigators reviewed the exclusion criteria for the taVNS pilot study in order to address these concerns.

##### Sub-Theme 3: Questions on Peripherals of Device

Participants asked a number of questions on the peripherals of the device such as mobile connectivity and wireless internet access (Wi-Fi) (e.g., “*And the phone gives you access to Wi- Fi, right? Because like if you don’t have Wi-Fi or you don’t**…you know what I mean, then it**…it’s supplying the Wi-Fi so you’re not going to*”), as well as accurate setup/connection of the device (e.g., “*I have a question. It’s a little strange, but I’m not a technology guy really, and I’m looking at this when**…when they’re showing**…explaining it, and I had asked XX (study investigator) about do you hear anything, how do you know if you set it up right? Because if you don’t hear it or feel it, how do you know, you**…you set it up right?*”), and additional devices (e.g, “*So, you’ll get an app on you existing phone or is it a separate device**…separate device? Free phone too, right?*”). It was also noted during the focus group that instructions can be included in other languages (e.g., Spanish) so that participants can select their preferred language during use. 

#### 3.2.3. Theme 3: Feedback on Research Methodology

The final theme that emerged focused on the participants’ feedback regarding adjustments to the research methodology. Participants made suggestions to improve the screening process, provided feedback on confidentiality concerns, and mentioned possible difficulties with the questionnaires. The participants also provided insight into potential challenges with participation and made some recommendations as to how to overcome those challenges and improve participation.

##### Sub-Theme 1: Adjustments to or Clarification of Screening Process

The main concern regarding screening for pilot study inclusion was with respect to having to still undergo additional PTSD screening despite a current diagnosis, who would have access to the screening information, and the location of screening for participation. For example, participants responded, “*I’m glad that you mentioned about**…about this**…this, you know, determining if there is true PTSD or no. In my case, I was certified, I really do, you still go through that process again?*” and “*Now, is the screening get**…given to somebody or like is it**…is there anybody have access to this screening?*” In terms of location, one participant stated “*For, me XX (town) would be easier because I’m in North**…in**…in XX (town), my work stops….especially, there is parking over there…parking is a big thing*.”

##### Sub-Theme 2: Confidentiality Concerns

Participants expressed concerns regarding their confidentiality in terms of both whether the device was tracking their private information and individuals outside of the study group having access to their information. With respect to privacy concerns using the device, one participant indicated that it is important to let potential participants know that the investigators will be able to see device usage, but that is the limit of what will be recorded from the device, “*That say,* “*Hey, you know, we’ll know,*” *but not like Big Brother is watching*.” While another participant followed up with “*like how**…you know, I would want to know like what have you known. I’m using it and how do you know I’m not using it three times a day and the**…you know, somehow address that*.” Participants also wanted to know who else had access to their information. One participant asked, “*So, is it a study again doesn’t go into the central?*” with central referring to the health care team from the Queens WTCHP and larger healthcare system. Once the facilitator reassured the participant that only the research team would have access to their information, the participant responded: “*That’s**…that’s actually really good*.” Additionally, the facilitator indicated that participation in the study would not impact their care, and another participant noted, “*…So, that’s important to tell people who maybe don’t tell the whole truth during a regular screening because they have other issues that …So, this might enable them to get help which is a great thing because it might enable them to get help without having to disclose*.”

##### Sub-Theme 3: Difficulties with Questionnaires

The idea of repeated survey measures at two points (some of which are also completed during their regular WTCHP annual visits) were of concern among the participants, with one saying, “*But you’re putting on the same questions over and over?*” and another expressing distaste and saying, “*Oh, I hate them [questionnaires]….this gives me PTSD, right now*.”

##### Sub-Theme 4: Challenges to Participation

Investigators inquired about challenges or barriers that WTC responders may face in terms of their participation within the study. Some responses included questioning veracity on self-report measures in that participants may be hesitant to participate if they need to be open about their symptoms, time constraints (e.g., “*I think it’s the typical like whatever we have said. It’s you know, time, there’s always a time, making it a priority*”), and medical issues and/or comorbidities that might affect inclusion criteria (e.g., “*I’m sorry, what**…what Participant 2 had mentioned too, medical issues, if you have preexisting medical issues, then people might feel certain trepidation that, ‘Oh, this could exacerbate what I have or I**…maybe I’m exempt from it,’ perhaps not warranted but just human nature that is so*”). Honesty around device usage was a concern cited by a participant, who stated: “*I think the biggest thing is to have them give you honest feedback. So, if they haven’t been using it, don’t lie and say you’ve used it.*”

##### Sub-Theme 5: Recommendations for Increasing Participation in Pilot Study

Investigators were also interested in obtaining feedback regarding ways to maximize pilot study participation among WTC responders such as being clear and upfront about compensation (e.g., “*So, I just feel like it’s five visits, you’ll get compensated a $150….Broken up and then it’s better than saying 25, 20, you know what I mean. Just say…I would say overall and then you break it down….So that**…this is the time commitment over three months or whatever it is. And then like 15 min a day, you know, and here’s the breakdown and you’ll get $150 for your, you know, for your time and**…and then you break it down. Then, I think that sounds different*.”) and also reducing notification fatigue (e.g., “*Notification fatigue is a real thing. That’s**…that’s**…the reason why people have hundreds of text messages, they don’t read (chuckles), but it’s just one of those things that you set, you know.*”)

Participants also indicated some positive aspects of the study that would serve to maximize study retention, such as ability to use the device anywhere (e.g., “*so now ever since COVID**…So, I have my**…my counseling on the phone. Because we weren’t able to go in and that was huge, huge, huge thing that you could do wherever….like I could do it on a lunch break, I can, you know, you can really do it anywhere and that’s huge, huge*”) and the perceived ease of compensation (e.g., “*Well, I think people are okay with having a card that they can**…they can use it just like any other credit card*”). Lastly, the participants were also cautioned to be mindful about the age group of the study participants when implementing and designing parts of the study given the potential for older adults to have greater difficulty with technology (e.g., “*And it’s just remembering that you’re dealing with an older population*”).

## 4. Discussion

It remains challenging to implement current clinical practice guidelines for the management of PTSD among the WTC responder population [22]. Previous work has shown the safety and effectiveness of VNS as an adjunct therapy to treat mental health disorders, showing the potential of using noninvasive VNS methods to address PTSD symptoms. In this study, a focus group of WTC responders was able to provide insight regarding the mental health needs of this responder population as well as the acceptability of using taVNS to treat elevated PTSD symptoms.

Three major themes emerged from the qualitative analysis of the focus group data. The first theme that arose involved the mental health needs of WTC responders with PTSD, with subthemes addressing barriers and potential solutions to facilitate receiving care. Participants unanimously agreed that current practices of psychotropic medications and talk therapy were not consistently beneficial, and barriers included transportation and access to care. Participants indicated that other mental health challenges have developed in conjunction with PTSD symptoms, and they often cannot be sufficiently managed due to societal stigma and perceived lack of interest and understanding from employers and the government.

The second theme was regarding the taVNS device, and subthemes included comments about the design, use, and physiological interactions. Participant reactions were extremely positive about the design of the device, and the novelty of a noninvasive and portable earpiece was very exciting. The focus group participants were enthusiastic about the potential of a new type of therapy, ease of use, and convenience of using the device anywhere and anytime. There were some concerns about the mechanisms of nerve stimulation and their effect on existing illnesses and comorbidities, and participants were assured of the safety of the device and potential benefits towards alleviating other symptoms. This discussion also reinforced the need to be very clear with potential participants about inclusion/exclusion criteria and the reasoning behind them in order for participants not to feel disappointed if they did not qualify for participation. Further, based on focus group participant feedback, the study team is making adjustments to peripherals, including adding subtitles to the instructional video, making the video available in Spanish, and ensuring that the instructional written materials and video are easily understood and explained slowly and in plain language for anyone to understand. In addition, the technology team is working on creating a system in which the device does not need to be connected to a strong Wi-Fi connection in order to properly provide therapy. During the next phase of the study, which is the RCT pilot trial, there will be an opportunity for participants to provide additional feedback regarding the device and peripherals in order to further enhance usability and acceptability for a future large trial.

The third theme concerned the RCT pilot study structure and feedback to maximize active participation. Subthemes identified concerns of the targeted patient population, particularly privacy and confidentiality, challenges to participation, and recommendations to increase participation and reduce attrition. The provision of confidentiality protection is crucial for potential patients and may also encourage participation, assuring participants that their privacy remains intact and will hopefully also encourage honesty in responses to study measures. Confidentiality language suggested by participants will be incorporated into the discussions with potential participants. Some additional potential challenges included time constraints and medical issues that may prevent participation. The study team is committed to working around participants’ schedules and meeting them in a location that is most convenient for them for the screening process and will use the feedback from the focus group participants to clearly explain the compensation structure and that it will also go towards transportation costs. Further, the focus group also emphasized the need for additional support for WTC responders who may not be eligible to participate in order to navigate and address any feelings of rejection. These suggestions will all be incorporated into the study methodology including clarifications around privacy.

The findings from this focus group of WTC responders highlight both the need for additional treatment options to reduce PTSD symptoms and the excitement about using a noninvasive, portable, and easy-to-use bioelectronic medicine device for a population that experiences barriers to access and adherence to PTSD treatment. TaVNS has successfully treated refractory epilepsy [23,24,25] and shown promising results in patients with inflammatory conditions [26], pre-diabetes [27], tinnitus [28], depression [29], oromotor dysfunction [30], rheumatoid arthritis [31], and stroke [32,33]. Previous studies suggest that targeting the vagus nerve for stimulation can suppress inflammation, modulate activity in mood-related areas of the brain, and impact sympathetic tone while reducing anxiety and other PTSD-associated symptoms [13,18,19]. Additionally, because the stimulation system can be controlled by a smartphone app, an early randomized trial of taVNS to treat depression found that patients could apply the unit themselves and did not need a healthcare professional nor clinical setting to administer therapy [34]; this makes it simple to use at home, relieving any stress of transportation or facility access. While participants were concerned about time constraints, it should be noted that taVNS therapy is applied for only 15 min per day at any preferred location for the patient. A future RCT can apply this novel therapeutic approach developed by noninvasive bioelectronic medicine technology while quantifying autonomic nervous system function to provide an option other than pharmacological- and psychotherapy-based therapies [35].

The current study has several limitations. First, the results are based on one focus group of six WTC responders with PTSD that were selected using criterion sampling, based on eligibility criteria and convenience sampling based on participant availability. As such, the participants did not represent the entire population of WTC responders living with PTSD, increasing the likelihood that their experiences with mental health treatment were not fully representative of the population. We also cannot rule out self-selection bias as a major limitation, such that the patients who participated may have differed from those who decided not to participate, since we did not fully assess the characteristics of those who chose not to participate or their reasons for not attending. One potential drawback in focus group discussions is the lack of guarantee that all those recruited will attend the discussion. Although we recruited 10 participants, only 6 ended up coming to the focus group, and thus the other 4 were considered “no shows”. It is generally accepted that between six and eight participants are sufficient to gain a variety of perspectives and not small enough not to become disorderly or fragmented [36]. The smaller sample size also increases the possibility that additional recommendations regarding the study methodology and device may have been missed, although by keeping the number of participants in the focus group small by design, we were able to stimulate all participants to participate actively in the discussion. The taVNS device has already undergone extensive beta testing and has been used with a variety of patient populations [37], which hopefully also serves to minimize issues around usability and acceptability. Further, the use of randomization in participant selection helped to ensure some variability in the demographics of those who participated in the focus group. Second, it is possible that there was a social desirability bias present during the focus group, whereby participants were more likely to speak positively about the device and methodology in order to appear more “socially acceptable” to the study team or perhaps mistakenly feel that this would increase their chances of being asked to participate in the pilot study. Social desirability issues are not uncommon in qualitative research [38]. However, the honesty with which the participants all spoke about their mental health challenges may have represented a somewhat reduced tendency towards this bias as it is typically less socially desirable to discuss psychological difficulties.

In conclusion, the current study underscores the continued need to develop novel approaches to treat PTSD among WTC responders, as it is clear that many responders are still searching for effective, accessible treatments even 20 years after the terrorist attacks on September 11, 2001. Further, findings highlighted that the intervention was assessed quite positively by focus group participants and that the taVNS device, as well as the pilot study methodology, will need few adaptations in order for the device and the study to be feasible and acceptable for use within this population. It is imperative that researchers and clinicians begin to investigate alternative treatments to reduce PTSD symptoms and perhaps augment existing EBTs for PTSD in order to meet the mental health needs of this group of individuals who risked their own lives by responding to the attacks on the WTC 20 years ago. This study represents a first, critical step in that process.

## Figures and Tables

**Figure 1 ijerph-19-04847-f001:**
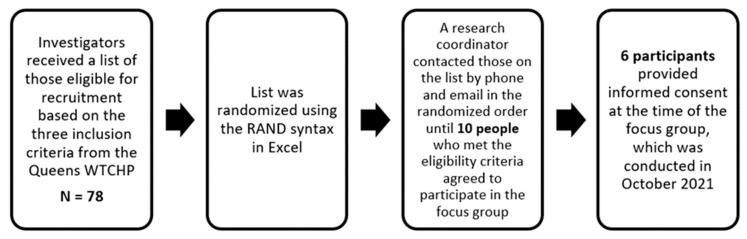
taVNS Focus Group Recruitment Process.

**Table 1 ijerph-19-04847-t001:** Identified Themes and Subthemes from Qualitative Analysis.

Theme 1: Mental Health Needs; Barriers/Facilitators to Engaging in Mental Health Care
*Subthemes* Mental health difficulties continue to be prevalent Systemic challenges to receiving access to treatment for PTSD and other mental health problems Suggestions to overcome barriers to mental health treatment
**Theme 2: Device Feedback**
*Subthemes* Initial reactions Concerns about interactions with other illnesses or comorbidities Questions on peripherals of device
**Theme 3: Feedback on Research Methodology**
*Subthemes* Adjustments to or clarification of screening process Confidentiality concerns Difficulties with questionnaires Challenges to participation Recommendations for increasing participation

## Data Availability

Completely de-identified transcripts can be made available upon request, but given the sensitive nature of qualitative data it would require further data sharing authorizations between institutions.

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
