# Peer review of "Understanding Mental Health Needs and Gathering Feedback on Transcutaneous Auricular Vagus Nerve Stimulation as a Potential PTSD Treatment among 9/11 Responders Living with PTSD Symptoms 20 Years Later: A Qualitative Approach"

_ijerph, 2022, doi:10.3390/ijerph19084847_

Round 1

Reviewer 1 Report

Manuscript Review IJERPH 1594842

This is a methodologically very sound and thoughtfully crafted manuscript that will have impact in the field of PTSD surrounding WTC responders. The pitfalls and limitations section is well-articulated as is the general layout of the Sections. There were minor problems with the referencing alignment in the text and some problems with the formatting that will certainly be addressed before the manuscript is published. This is a small but important contribution to research into additional forms of treatment for PTSD in 9/11 responders. The only wish is that in future manuscripts, since there are literally hundreds of Traditiona, Complementary and Integrative Medicine and Health Therapies out there (well-outlined by the World Health Organizations Section on Traditional, Complementary and Integrative Medicine and Health (TCIM), that they spend some time exploring what other modalities might be available to treat PTSD in 9/11 responders, be they mind/body therapies, natural products/herbal formulations, manipulative therapies, movement therapies and Traditional Medicines (Ayuvedic Medicine, Traditional Chinese Medicine). We know that almost 40% of all Americans seek alternative treatments to pharmaceuticals and surgery for a variety of conditions, so it might be beneficial to spend a few paragraphs on what else has been looked at the VA, for example, for Veterans suffering from deployment-related and war-related PTSD.   

Materials and Methods

Under Qualitative Analysis:

Please verify that the references are corrected aligned in the text. For example, reference 20 in the below paragraph does not correspond to Braun and Clarke (Braune and Clarke are reference 21). Also the Braun and Clarke reference is listed in the reference as reference 21 and reference 22—please make sure this is all corrected.

 “The focus group discussion was audio-recorded and professionally transcribed verbatim. An inductive thematic analysis [20] was used to analyze the transcribed focus group discussion. To improve credibility, dependability and transferability of results, peer de-briefing, documentation of results, and researcher triangulation were employed, following the six phases set forth by Braun and Clarke [20].”

Please make sure formatting is corrected as in below where the subject header, “Theme 3: Feedback on Research Methodology,” has been placed at the end of a paragraph and not separated.

“It was also noted during the focus group that instructions can be included in other languages (e.g., Spanish) so that participants can select their preferred language during use. Theme 3: Feedback on Research Methodology”

Under: Subtheme 1: Adjustments to or clarification of screening process

Replace “In terms of location, on participant” with “In terms of location, one participant”

Under Sub-Theme 5:

Make sure quotations are all in italics please:

“e.g., “So, I just feel like it's five visits, you'll get compensated a $150….Broken up and then it's better than saying 25, 20, you know what I mean. Just say --I would say overall and then you break it down….So that -- this is the time commitment over three months or whatever it is. And then like 15 minutes a day, you know, and here's the breakdown and you'll get $150 for your, you know, for your time and -- and then you break it down. Then, I think that sounds different.”) and also reducing notification fatigue (e.g., “Notification fatigue is a real thing. That's -- that's -- the reason why people have hundreds of text messages, they don’t read (chuckles), but it's just one of those things that you set, you know.)”

Author Response

Point 1: This is a methodologically very sound and thoughtfully crafted manuscript that will have impact in the field of PTSD surrounding WTC responders. The pitfalls and limitations section is well-articulated as is the general layout of the Sections. There were minor problems with the referencing alignment in the text and some problems with the formatting that will certainly be addressed before the manuscript is published. This is a small but important contribution to research into additional forms of treatment for PTSD in 9/11 responders. The only wish is that in future manuscripts, since there are literally hundreds of Traditiona, Complementary and Integrative Medicine and Health Therapies out there (well-outlined by the World Health Organizations Section on Traditional, Complementary and Integrative Medicine and Health (TCIM), that they spend some time exploring what other modalities might be available to treat PTSD in 9/11 responders, be they mind/body therapies, natural products/herbal formulations, manipulative therapies, movement therapies and Traditional Medicines (Ayuvedic Medicine, Traditional Chinese Medicine). We know that almost 40% of all Americans seek alternative treatments to pharmaceuticals and surgery for a variety of conditions, so it might be beneficial to spend a few paragraphs on what else has been looked at the VA, for example, for Veterans suffering from deployment-related and war-related PTSD.  

Response: Thank you for your comments. We have adjusted the reference alignment and formatting. We have also incorporated a section in the introduction that includes alternative treatments for PTSD including treatment options examined by the VA.

 Point 2: Materials and Methods

Under Qualitative Analysis:

Please verify that the references are corrected aligned in the text. For example, reference 20 in the below paragraph does not correspond to Braun and Clarke (Braune and Clarke are reference 21). Also the Braun and Clarke reference is listed in the reference as reference 21 and reference 22—please make sure this is all corrected.

 “The focus group discussion was audio-recorded and professionally transcribed verbatim. An inductive thematic analysis [20] was used to analyze the transcribed focus group discussion. To improve credibility, dependability and transferability of results, peer de-briefing, documentation of results, and researcher triangulation were employed, following the six phases set forth by Braun and Clarke [20].”

Response: All references were double checked to ensure alignment with in-text citations and the references section.

Point 3: Please make sure formatting is corrected as in below where the subject header, “Theme 3: Feedback on Research Methodology,” has been placed at the end of a paragraph and not separated.

“It was also noted during the focus group that instructions can be included in other languages (e.g., Spanish) so that participants can select their preferred language during use. Theme 3: Feedback on Research Methodology”

Under: Subtheme 1: Adjustments to or clarification of screening process

Replace “In terms of location, on participant” with “In terms of location, one participant”

Under Sub-Theme 5:

Make sure quotations are all in italics please:

“e.g., “So, I just feel like it's five visits, you'll get compensated a $150….Broken up and then it's better than saying 25, 20, you know what I mean. Just say --I would say overall and then you break it down….So that -- this is the time commitment over three months or whatever it is. And then like 15 minutes a day, you know, and here's the breakdown and you'll get $150 for your, you know, for your time and -- and then you break it down. Then, I think that sounds different.”) and also reducing notification fatigue (e.g., “Notification fatigue is a real thing. That's -- that's -- the reason why people have hundreds of text messages, they don’t read (chuckles), but it's just one of those things that you set, you know.)”

Response: Edited as suggested. Formatting was corrected around the heading for Theme 3. Typos and formatting around quotations have also been addressed in Subthemes 1 and 5 respectively.

Reviewer 2 Report

The authors have conducted a qualitative study to elucidate barriers and facilitators to engagement in mental health care and the feasibility and acceptability of using the taVNS device as a treatment for PTSD symptoms among responders of WTC terrorist attacks.   Authors have clarified three themes in conducting the research, Mental Health needs; barriers/facilitators to engaging in mental health care, Device Feedback, and Feedback on Research Methodology.   The Manuscript is well written and elaborates taVNS as an important treatment options for responders suffering from PTSD.

Author Response

Point 1: The authors have conducted a qualitative study to elucidate barriers and facilitators to engagement in mental health care and the feasibility and acceptability of using the taVNS device as a treatment for PTSD symptoms among responders of WTC terrorist attacks.  Authors have clarified three themes in conducting the research, Mental Health needs; barriers/facilitators to engaging in mental health care, Device Feedback, and Feedback on Research Methodology. The Manuscript is well written and elaborates taVNS as an important treatment options for responders suffering from PTSD.

Response: Thank you for your comments.

Reviewer 3 Report

This is a well-planned and executed study, although involving only 6 participants. Including the 'Discussion Guide' as supplementary material is very helpful and contributes positively to the organization of the study and manuscript.

Three main themes with their subthemes were appropriately presented and discussed in order to better plan a subsequent RCT. The participants' perspectives in identifying mental health needs and barriers and facilitators to mental health care engagement were meticulously catered for. Also, the acceptability (drug-free therapy option, guided by technological innovation) and possible difficulties in the implementation of therapy (fitting) with the taVNS device were appropriately addressed. 

Hopefully, this study should lead to a more comprehensive subsequent study.

Author Response

Point 1: This is a well-planned and executed study, although involving only 6 participants. Including the 'Discussion Guide' as supplementary material is very helpful and contributes positively to the organization of the study and manuscript.

Response: Thank you for your comments. 

Point 2: Three main themes with their subthemes were appropriately presented and discussed in order to better plan a subsequent RCT. The participants' perspectives in identifying mental health needs and barriers and facilitators to mental health care engagement were meticulously catered for. Also, the acceptability (drug-free therapy option, guided by technological innovation) and possible difficulties in the implementation of therapy (fitting) with the taVNS device were appropriately addressed. Hopefully, this study should lead to a more comprehensive subsequent study.

Response: Thank you so much for your comments. We wanted to note that we made very minor edits to the layout of the themes in Table 1, for an easier read.

Reviewer 4 Report

PTSD remains one of the major mental health disorders in WTC responders. Potential non-invasive new treatment for PTSD is of great importance for WTC responders. The topic of the study discussed in the manuscript is of importance and interest. However, it seems to me, the manuscript is poorly written with a serious lack of scientific rigor. The description of the study design and method used is not clear enough to be informative. It might be helpful if the authors provide a figure or diagram that outlines the the number of subjects in each step of selection process. For example, how many people are eligible to be selected? how many contacted, what can you say about the 6 finally selected  out of the 10 consented subjects? how can you exclude possibilities of self-selection and other biases for subjects to consent and to actually participate?  Is there any initial consideration for sample size and statistical power or statistical uncertainty associated with this  study? With possible self-selection and other biases on the 6 out of 10 participants, as is well known, when randomly tossing 10 coins, six of the 10 tosses appear to be head or six of the tosses appear to tails can occur with quite high probabilities. Thus, it is inconclusive to non-convincing at all to say anything about consensus or pattern based on 6 subjects in this context. More careful discussion would be needed. Also, with only 6 subjects in the study, the subject-specific analysis should also be reported. Is there any evidence that some of the 6 subjects might have strong and influential opinions to the data sets? If there any difference between male and female subjects? Without only 6 subjects, can you really identify so many themes and sub-themes with reasonable certainty? Are there missing answers or missing data for some of the questions? Summarizes in qualitative analysis typically need to provide basic information including the number of answers for each item that the consensus or summary points were based on. It seems, the medical device has been used in other context and in a large number of publications, why the authors do not consider using some published data to support the current study to reduce the severe negative impact of the small sample size of the current study? Or, the authors might combine the current data with some existing survey on WTC responders to overcome the limitation of sample size. The authors can also conduct a small-scale survey to complement the current study if no existing survey data can be used. It seems to me, the limitation of sample size of the current study, the potential impact of selection bias and other biases, the lack of uncertainty considerations need to be addressed in the description of the study design and as major limitations discussed in the Discussion section.

Author Response

Point 1: PTSD remains one of the major mental health disorders in WTC responders. Potential non-invasive new treatment for PTSD is of great importance for WTC responders. The topic of the study discussed in the manuscript is of importance and interest. However, it seems to me, the manuscript is poorly written with a serious lack of scientific rigor. The description of the study design and method used is not clear enough to be informative. It might be helpful if the authors provide a figure or diagram that outlines the number of subjects in each step of selection process. For example, how many people are eligible to be selected? how many contacted, what can you say about the 6 finally selected  out of the 10 consented subjects?

Response: Thank you for your feedback. We have added a figure demonstrating the recruitment process for Aim 1 of our study, highlighting the total number of eligible participants, number recruited, and number of participants who actually showed to the focus group. All 78 participants were reached out to by email and 50 were reached by phone, and the first 10 participants who agreed to come in for the focus group were selected. We have also added a sentence regarding the PCL scores of our participants.

Point 2: how can you exclude possibilities of self-selection and other biases for subjects to consent and to actually participate?  Is there any initial consideration for sample size and statistical power or statistical uncertainty associated with this  study? With possible self-selection and other biases on the 6 out of 10 participants, as is well known, when randomly tossing 10 coins, six of the 10 tosses appear to be head or six of the tosses appear to tails can occur with quite high probabilities. Thus, it is inconclusive to non-convincing at all to say anything about consensus or pattern based on 6 subjects in this context. More careful discussion would be needed. Also, with only 6 subjects in the study, the subject-specific analysis should also be reported. Is there any evidence that some of the 6 subjects might have strong and influential opinions to the data sets? If there any difference between male and female subjects? Without only 6 subjects, can you really identify so many themes and sub-themes with reasonable certainty? Are there missing answers or missing data for some of the questions? Summarizes in qualitative analysis typically need to provide basic information including the number of answers for each item that the consensus or summary points were based on. It seems, the medical device has been used in other context and in a large number of publications, why the authors do not consider using some published data to support the current study to reduce the severe negative impact of the small sample size of the current study? Or, the authors might combine the current data with some existing survey on WTC responders to overcome the limitation of sample size. The authors can also conduct a small-scale survey to complement the current study if no existing survey data can be used. It seems to me, the limitation of sample size of the current study, the potential impact of selection bias and other biases, the lack of uncertainty considerations need to be addressed in the description of the study design and as major limitations discussed in the Discussion section.

Response: Since attending the focus group was completely voluntary, we cannot make any assumptions regarding the six participants that attended, except that they were interested in learning about a potential new treatment for PTSD which has never been used with this responder population. We were only able to include demographic data, based on what we received from the World Trade Center Health Program General Responder Data Center based on IRB protocols set in place as we were preparing for the focus group. We are aware of self-selection and other biases, which have been added to the discussion section. For the purpose of our focus group, we wanted a small, intimate group to serve as beta testers and provide their input and feedback using the questions from the discussion guide as a prompt. With a large-scale clinical trial in the future, we would look into potentially addressing the points above. We have included the limitations of our small sample size in the discussion section.

Reviewer 5 Report

First, I would like to thank the opportunity to read this study. This study adds to the literature by conducting a qualitative study with six responders from the Queens World Trade Center (WTC) Health Program who had elevated symptoms of posttraumatic stress disorder (PTSD). Although the number of participants may compromise the generalization of the results found in the present study, I found this study very interesting and very well structured. There are just some typos throughout the paper that will be easily amended by reading the paper carefully again. I have nothing substantial to add and I congratulate the authors for the study developed.

Author Response

Point 1: First, I would like to thank the opportunity to read this study. This study adds to the literature by conducting a qualitative study with six responders from the Queens World Trade Center (WTC) Health Program who had elevated symptoms of posttraumatic stress disorder (PTSD). Although the number of participants may compromise the generalization of the results found in the present study, I found this study very interesting and very well structured. There are just some typos throughout the paper that will be easily amended by reading the paper carefully again. I have nothing substantial to add and I congratulate the authors for the study developed.

Response: Thank you so much for your comments. Yes, we agree that the sample size is a limitation and have addressed this in the discussion section. We have done a thorough spell check to address the typos.

Reviewer 6 Report

Many thanks for giving me the chance to review this interesting article. Please find below some comments:

  • To my view this should be avoided in the title “and a potential new treatment”;
  • I suggest adding in the title the qualitative approach;
  • It’s not clear with is the aim of the study;
  • Methods should be more structured;
  • Introduction is very long. I suggest shortening it;
  • The aim of the study is not very clear in the introduction;
  • Please explain why 10 and not more or less. Did you reach data saturation with ten?
  • In qualitative studies “transferability of results” is very difficult so I suggest avoiding this or at least rephrasing;
  • Please explain why 4 didn’t participate?

Author Response

Many thanks for giving me the chance to review this interesting article. Please find below some comments:

Point 1:

  • To my view this should be avoided in the title “and a potential new treatment”;
  • I suggest adding in the title the qualitative approach;

Response: Thank you for your feedback. We have changed the title to “Understanding mental health needs and gathering feedback on Transcutaneous Auricular Vagus Nerve Stimulation as a potential PTSD treatment among 9/11 responders living with PTSD symptoms 20 years later: A qualitative approach”.

 Point 2:

  • It’s not clear with is the aim of the study; The aim of the study is not very clear in the introduction;

Response: The aims of the focus group and the study are highlighted in the last paragraph of the Introduction section. The aims are to understand responders’ perspectives on: 1) the mental health needs of WTC responders as well as barriers and facilitators to engagement in mental health care and 2) the feasibility and acceptability of using the taVNS device as a treatment for PTSD symptoms as well as the pilot study methodology.

Point 3:

  • Methods should be more structured;

Response: A figure was added to the methods section to add clarity around recruitment and section headings were slightly modified to add structure to the methods section.

Point 4:

  • Introduction is very long. I suggest shortening it;

Response: We removed a few sentences in the introduction but shortening it further would impact the clarity and flow of the paper.

Point 5:

  • Please explain why 10 and not more or less. Did you reach data saturation with ten?

Response: We felt that 10 participants would allow us to reach a reasonable consensus on the modifications that should be made for this particular population. The intervention has been beta tested with other populations, but we felt that 10 would add to the overall understanding of usability. Additional information was added to the discussion section on this point.

 Point 6:

  • In qualitative studies “transferability of results” is very difficult so I suggest avoiding this or at least rephrasing;

Response: Removed the phrase “transferability of results” as suggested.

Point 7:

  • Please explain why 4 didn’t participate.

Response: We are not sure why 4 participants did not attend the focus group after confirming that they would be there even the morning of the focus group. Two participants later emailed us citing work delays as reason for not attending the focus group. This is now included in the participant recruitment section.

Round 2

Reviewer 4 Report

The authors are very responsive to my comments and have addressed the major issues I raised.